# On dissipation time scales of the basic second-order moments: the effect on the Energy and Flux-Budget (EFB) turbulence closure for stably stratified turbulence

Evgeny Kadantsev[1,2], Evgeny Mortikov[3,4,5], Andrey Glazunov[4,3], Nathan Kleeorin[6,7], Igor Rogachevskii[6,8]

[1]Finnish Meteorological Institute, Helsinki, 00101, Finland
[2]Institute for Atmospheric and Earth System Research / Physics, Faculty of Science, University of Helsinki, 00014, Finland
[3]Lomonosov Moscow State University, 117192, Russia
[4]Institute of Numerical Mathematics, Russian Academy of Sciences, Moscow, 119991, Russia
[5]Moscow Center of Fundamental and Applied Mathematics, 117192, Russia
[6]Department of Mechanical Engineering, Ben-Gurion University of the Negev, P. O. B. 653, Beer-Sheva, 8410530, Israel
[7]Institute of Continuous Media Mechanics, Korolyov str. 1, 614013 Perm, Russia
[8]Nordita, Stockholm University and KTH Royal Institute of Technology, 10691 Stockholm, Sweden

*Correspondence to*: Evgeny Kadantsev (evgeny.kadantsev@helsinki.fi)

**Abstract.** The dissipation rates of the basic second-order moments are the key parameters controlling turbulence energetics and spectra, turbulent fluxes of momentum and heat, and playing a vital role in turbulence modelling. In this paper, we use the results of Direct Numerical Simulations (DNS) to evaluate dissipation rates of the basic second-order moments and revise the
20 energy and flux-budget turbulence closure theory for stably stratified turbulence. We delve into the theoretical implications of this approach and substantiate our closure hypotheses through DNS data. We also show why the concept of down-gradient turbulent transport becomes incomplete when applied to the vertical turbulent flux of potential temperature under stable stratification. We reveal essential feedback between the turbulent kinetic energy, the vertical turbulent flux of buoyancy and the turbulent potential energy, which is responsible for maintaining shear-produced stably stratified turbulence for any
Richardson number.

## 1 Introduction

Turbulence and associated turbulent transport have been studied theoretically, experimentally, observationally and numerically during decades [see books by Batchelor (1953); Monin and Yaglom (1971, 2013); Tennekes and Lumley (1972); Frisch (1995); Pope (2000); Davidson (2013); Rogachevskii (2021), and references therein], but some important questions remain. This is
30 particularly true in applications to atmospheric physics and geophysics where Reynolds and Peclet numbers are extremely

large so that the governing equations are strongly nonlinear. The classical Kolmogorov's theory (Kolmogorov 1941a,b; 1942; 1991) has been formulated for neutrally stratified homogeneous and isotropic turbulence.

In atmospheric boundary layers, temperature stratification causes turbulence to become anisotropic and inhomogeneous making some assumptions underlying Kolmogorov's theory questionable. Numerous alternative turbulence closure theories [see reviews by Weng and Taylor (2003); Umlauf and Burchard (2005); Mahrt (2014)] have been formulated using the budget equations not only for turbulent kinetic energy (TKE), but also for turbulent potential energy (TPE) (see, e.g., Holloway, 1986; Ostrovsky and Troitskaya, 1987; Dalaudier and Sidi, 1987; Hunt et al., 1988; Canuto and Minotti, 1993; Schumann and Gerz, 1995; Hanazaki and Hunt, 1996; Keller and van Atta, 2000; Canuto et al., 2001; Stretch et al., 2001; Cheng et al., 2002; 2002; Hanazaki and Hunt, 2004; Rehmann and Hwang, 2005; Umlauf, 2005). The budget equations for all three energies, TKE, TPE and total turbulent energy (TTE), were considered by Canuto and Minotti (1993), Elperin et al. (2002, 2006), Zilitinkevich et al. (2007), and Canuto et al. (2008).

The energy and flux budget (EFB) turbulence closure theory which is based on the budget equations for the densities of TKE, TPE and turbulent fluxes of momentum and heat, has been developed for stably stratified atmospheric flows (Zilitinkevich et al., 2007, 2008, 2009, 2013; Kleeorin et al. 2019), for surface layers in atmospheric convective turbulence (Rogachevskii et al. 2022) and the core of the convective boundary layer (Rogachevskii and Kleeorin, 2024), as well as for passive scalar transport (Kleeorin et al. 2021). The EFB closure theory has shown that strong atmospheric stably stratified turbulence is maintained by large-scale shear (mean wind) for any stratification, and the "critical" Richardson number, considered many years as a threshold between the turbulent and laminar states of the flow, actually separates two turbulent regimes: the strong turbulence typical of atmospheric boundary layers and the weak three-dimensional turbulence typical of the free atmosphere and characterised by a strong decrease in the turbulent heat transfer in comparison to the momentum transfer.

Some other turbulent closure models (Mauritsen et al. 2007, Canuto et al., 2008, Sukoriansky and Galperin, 2008, Li et al. 2016) do not imply the critical Richardson number, so shear-generated turbulent mixing may persist for any stratification. In particular, Mauritsen et al. (2007) have developed a turbulent closure based on the budget equation for TTE (instead of TKE) and different observational findings to take into account the mean flow stability. They used this turbulent closure model to study the turbulent transfer of heat and momentum under very stable stratification. In their model, whereas the turbulent heat flux tends toward zero beyond a certain stability limit, the turbulent stress stays finite. However, the model by Mauritsen et al. (2007) does not use the budget equation for TPE and the vertical turbulent heat flux.

L'vov et al. (2008) have performed detailed analyses of the budget equations for the Reynolds stresses in the turbulent boundary layer (relevant to the strong turbulence regime) taking explicitly into consideration the dissipative effect in the horizontal turbulent heat flux budget equation, in contrast to the EFB "effective-dissipation approximation" adopted in the EFB turbulent closure model. However, the theory by L'vov et al. (2008) still contains the critical gradient Richardson number for the existence of the shear-produced turbulence.

Sukoriansky and Galperin (2008) apply a quasi-normal scale elimination theory that is similar to the renormalization group analysis. Sukoriansky and Galperin (2008) do not use the budget equations for TKE, TPE and TTE in their analysis. This

theory correctly describes the dependence of the turbulent Prandtl number versus the gradient Richardson number and does not imply the critical gradient Richardson number for the existence of turbulence. However, this approach does not have detailed Richardson number dependences of the other non-dimensional parameters, like the ratio between TPE and TTE, dimensionless turbulent flux of momentum or dimensionless vertical turbulent flux of potential temperature. Their background non-stratified shear-produced turbulence is assumed to be isotropic and homogeneous. Canuto et al. (2008) have generalised their original model (see Cheng et al., 2002) introducing the new parameterization for the buoyancy time scale to accommodate the existence of stably stratified shear-produced turbulence at arbitrary Richardson numbers.

Li et al. (2016) have developed the co-spectral budget (CSB) closure approach which is formulated in the Fourier space and integrated across all turbulent scales to obtain turbulent characteristics in physical space. The CSB model allows turbulence to exist at any gradient Richardson number. However, the CSB model yields different predictions for the vertical anisotropy versus Richardson number compared to the EFB theory. All state-of-the-art turbulent closures follow the so-called Kolmogorov hypothesis: all dissipation time scales of turbulent second-order moments are assumed to be proportional to each other, which at first glance looks reasonable but, in fact, hypothetical for stably stratified turbulence.

The present study aims to demonstrate the dependence of dissipation time scales of basic second-order moments on stability through DNS experiments. The obtained numerical results allow us to modify the EFB turbulence closure theory to account for that dependency. It is worth noting that the DNS presented here are limited to bulk Richardson numbers (based on the wall velocity and temperature differences and channel height) up to $Ri_b = 0.11$ and Reynolds numbers (based on the wall velocity difference and channel height, see Sect. 3) up to $Re = 120000$.

This paper is organised as follows. In Section 2, we formulate basic budget equations and main assumptions in the framework of the EFB turbulence closure theory. Section 3 describes the setup for DNS of stably stratified turbulent plane Couette flow to determine the vertical profiles of the dissipation time scales of turbulent second-order moments. In Section 4, we formulate the modified EFB turbulence closure theory considering the dependencies of the dissipation time scales of basic second-order moments on the gradient Richardson number obtained from DNS. There, we also perform validation of the modified EFB turbulence closure model which yields vertical profiles of the basic turbulence parameters (including the turbulent Prandtl number, the ratio of TPE to TKE, the normalised turbulent heat flux, etc.) using the data from the DNS. Finally, in Section 5, we discuss the obtained results and draw the conclusions.

## 2 Problem setting and basic equations

We consider plane-parallel, stably stratified dry-air flow and employ the familiar budget equations underlying turbulence-closure theory (e.g., Kleeorin et al. 2021; Zilitinkevich et al., 2013; Kaimal and Fennigan, 1994; Canuto et al., 2008) for the Reynolds stress, $\tau_{ij} = \langle u_i u_j \rangle$, the turbulent flux of potential temperature, $F_i = \langle \theta u_i \rangle$, and the intensity of potential temperature fluctuations, $E_\theta = \langle \theta^2 \rangle / 2$:

$$\frac{D\tau_{ij}}{Dt} + \frac{\partial}{\partial z}\Phi_{ij3}^{(\tau)} = -\tau_{i3}\frac{\partial U_j}{\partial z} - \tau_{j3}\frac{\partial U_i}{\partial z} - \left[\varepsilon_{ij}^{(\tau)} - \beta\left(F_j\delta_{i3} + F_i\delta_{j3}\right) - Q_{ij}\right], \tag{1}$$

$$\frac{DF_i}{Dt} + \frac{\partial}{\partial z}\Phi_i^{(F)} = \beta\delta_{i3}\langle\theta^2\rangle - \frac{1}{\rho_0}\langle\theta\frac{\partial p}{\partial x_i}\rangle - \tau_{i3}\frac{\partial\Theta}{\partial z} - F_z\frac{\partial U_i}{\partial z} - \varepsilon_i^{(F)}, \tag{2}$$

$$\frac{DE_\theta}{Dt} + \frac{\partial}{\partial z}\Phi^{(\theta)} = -F_z\frac{\partial\Theta}{\partial z} - \varepsilon_\theta. \tag{3}$$

Here, $x_1 = x$ and $x_2 = y$ are horizontal coordinates, $x_3 = z$ is the vertical coordinate; $t$ is time; $\mathbf{U} = (U_1, U_2, U_3) = (U, V, W)$ is the mean flow velocity; $\mathbf{u} = (u_1, u_2, u_3) = (u, v, w)$ are velocity fluctuations; $\Theta = T(P_0/P)^{1-1/\gamma}$ is the mean potential temperature (expressed through absolute temperature, $T$, and pressure, $P$); $T_0$, $P_0$ and $\rho_0$ are reference values of temperature, pressure and density, respectively; $\gamma = c_p/c_v = 1.41$ is the ratio of specific heats; $\theta$ and $p$ are fluctuations of potential temperature and pressure; $D/Dt = \partial/\partial t + U_k\partial/\partial x_k$ is the advective derivative; angle brackets denote averaging; $\beta = g/T_0$ is the buoyancy parameter; $g$ is the acceleration due to gravity; $\delta_{ij}$ is the unit tensor ($\delta_{ij} = 1$ for $i = j$ and $\delta_{ij} = 0$ for $i \neq j$); $\Phi_{ij3}^{(\tau)}$, $\Phi_i^{(F)}$ and $\Phi^{(\theta)}$ are the third-order moments, which describe turbulent transport of the second-order moments under consideration:

$$\Phi_{ij3}^{(\tau)} = \langle u_i u_j w\rangle + \frac{1}{\rho_0}\left(\langle pu_i\rangle\delta_{j3} + \langle pu_j\rangle\delta_{i3}\right) - \nu\left(\langle u_i\frac{\partial u_j}{\partial z}\rangle + \langle u_j\frac{\partial u_i}{\partial z}\rangle\right), \tag{4}$$

$$\Phi_i^{(F)} = \langle u_i w\theta\rangle - \nu\langle\theta\frac{\partial u_i}{\partial z}\rangle - \kappa\langle u_i\frac{\partial\theta}{\partial z}\rangle, \tag{5}$$

$$\Phi^{(\theta)} = \frac{1}{2}\langle\theta^2 w\rangle - \frac{\kappa}{2}\frac{\partial}{\partial z}\langle\theta^2\rangle; \tag{6}$$

$Q_{ij}$ are the correlations between fluctuations of pressure and strain-rate tensor, which control the interactions between the Reynolds stress components:

$$Q_{ij} = \frac{1}{\rho_0}\langle p\left(\frac{\partial u_i}{\partial x_j} + \frac{\partial u_j}{\partial x_i}\right)\rangle. \tag{7}$$

Here, $\varepsilon_{ij}^{(\tau)}$, $\varepsilon_i^{(F)}$ and $\varepsilon_\theta$ are the dissipation rates of the second-order moments:

$$\varepsilon_{ij}^{(\tau)} = 2\nu\langle\frac{\partial u_i}{\partial z}\frac{\partial u_j}{\partial z}\rangle, \tag{8}$$

$$\varepsilon_i^{(F)} = (\nu + \kappa)\langle\frac{\partial u_i}{\partial z}\frac{\partial\theta}{\partial z}\rangle, \tag{9}$$

$$\varepsilon_\theta = \kappa\langle\left(\frac{\partial\theta}{\partial z}\right)^2\rangle, \tag{10}$$

where $\nu$ is kinematic viscosity and $\kappa$ is thermal conductivity.

The budgets of TKE components, $E_i = \langle u_i^2\rangle/2$ ($i = 1,2,3$), are determined by Eq. (1) for $i = j$, which yields the familiar TKE budget equation:

$$120 \quad \frac{DE_K}{Dt} + \frac{\partial}{\partial z}\left(\frac{1}{2}\langle u_i^2 w\rangle + \frac{1}{\rho_0}\langle pw\rangle - \frac{\nu}{2}\frac{\partial\langle u_i^2\rangle}{\partial z}\right) = -\boldsymbol{\tau}\cdot\frac{\partial\mathbf{U}}{\partial z} + \beta F_z - \varepsilon_K, \tag{11}$$

where $E_K = \sum E_i$ is TKE and $\varepsilon_K = \sum \varepsilon_{ii}^{(\tau)}/2$ is the TKE dissipation rate. The sum of the terms $Q_{ii}$ (the trace of the tensor $Q_{ij}$) is equal to zero because of the incompressibility constraint on the flow velocity field, $\partial u_i/\partial x_i = 0$, i.e. $Q_{ij}$ only redistribute energy between TKE components.

Likewise, $\varepsilon_\theta$ is the dissipation rate of the intensity of potential temperature fluctuations, $E_\theta$; and $\varepsilon_i^{(F)}$ are the dissipation rates
of the three components of the turbulent flux of potential temperature, $F_i$.

Following Kolmogorov (1941, 1942), the dissipation rates $\varepsilon_K$ and $\varepsilon_\theta$ are taken proportional to the dissipating quantities divided by corresponding time scales,

$$\varepsilon_K = \frac{E_K}{t_K}, \varepsilon_\theta = \frac{E_\theta}{t_\theta}, \tag{12}$$

where $t_K$ is the TKE dissipation time scale and $t_\theta$ is the dissipation time scale of $E_\theta$. Here, the formulation of the dissipation
rates is not hypothetical: it merely expresses one unknown (dissipation rate) through another (dissipation time scale).

In this study, we consider the EFB theory in its simplest, algebraic form, neglecting non-steady terms in all budget equations and neglecting divergence of the fluxes of TKE, TPE and fluxes of $F_z$ (determined by third-order moments). This approach is reasonable because, e.g., the characteristic times of variations of the second moments are much larger than the turbulent time scales for large Reynolds and Peclet numbers. We also assume that the terms related to the divergence of the fluxes of TKE
and TPE for stably stratified turbulence are much smaller than the rates of production and dissipation in budget equations (3) and (11). In this case, the TKE budget equation, Eq. (11), and the budget equation for $E_\theta$, Eq. (3), become

$$0 = -\tau\frac{\partial U}{\partial z} + \beta F_z - \varepsilon_K, \tag{13}$$

$$0 = -F_z\frac{\partial\Theta}{\partial z} - \varepsilon_\theta. \tag{14}$$

The intensity of the potential temperature fluctuations $E_\theta$ determines TPE:

$$140 \quad E_P = \frac{\beta E_\theta}{\partial\Theta/\partial z}, \tag{15}$$

so that Eq. (14) becomes

$$0 = -\beta F_z - \varepsilon_P, \tag{16}$$

Where $\varepsilon_P = E_P/t_\theta$ is the TPE dissipation time.

The first term on the right-hand side (r.h.s.) of Eq. (13), $-\tau\,\partial U/\partial z$, is the rate of the TKE production, while the second term,
$\beta F_z$, is the buoyancy which in stably stratified flow causes decay of TKE, i.e., it results in conversion of TKE into TPE. The ratio of these terms is the flux Richardson number:

$$\mathrm{Ri}_f \equiv -\frac{\beta F_z}{\tau \partial U/\partial z}, \tag{17}$$

and this dimensionless parameter characterises the effect of stratification on turbulence.

Taking into account Eq. (17), the steady-state versions of TKE and TPE budget equations, Eqs. (13) and (14), can be rewritten as

$$E_K = \tau \frac{\partial U}{\partial z}\left(1 - \mathrm{Ri}_f\right) t_K, \tag{18}$$

$$E_P = \tau \frac{\partial U}{\partial z}\mathrm{Ri}_f t_\theta. \tag{19}$$

Thus, the ratio of TPE to TKE is:

$$\frac{E_P}{E_K} = \frac{\mathrm{Ri}_f}{1-\mathrm{Ri}_f}\frac{t_\theta}{t_K}. \tag{20}$$

Zilitinkevich et al. (2013) suggested the following relation linking $Ri_f$ with another stratification parameter, $z/L$:

$$\mathrm{Ri}_f = \frac{kz/L}{1+kR_\infty^{-1}z/L}, \qquad \frac{z}{L} = \frac{R_\infty}{k}\frac{\mathrm{Ri}_f}{R_\infty-\mathrm{Ri}_f}, \tag{21}$$

where $L = -\tau^{3/2}/\beta F_z$ is the Obukhov length scale, $k = 0.4$ is the von Kármán constant, and $R_\infty = 0.2$ is the maximum value of the flux Richardson number.

On the r.h.s. of Eq. (20), there is an unknown ratio of two dissipation time scales, $t_\theta/t_K$. The Kolmogorov hypothesis suggests that it is a universal constant. We do not imply this assumption, but instead investigate a possible stability dependency of dissipation time scales ratios and improve the EFB turbulence closure model accounting for it. To this end, we perform DNS of stably stratified turbulent plane Couette flow (see Section 3) to measure the dissipation time scales of basic second-order moments and validate the modified EFB turbulence closure model (see Section 4).

## 3 Methods

For our study, we conducted a series of direct numerical simulations of stably stratified turbulent plane Couette flow. This flow occurs between two parallel plates that move relative to each other, producing shear and turbulence, with the plates having different temperatures, thus creating stable stratification. In Couette flow, the total (turbulent plus molecular) vertical fluxes of momentum and potential temperature remain constant, independent of distance from the walls, which, in particular, assures a very certain fixed value of the Obukhov length scale. Fig. 1 illustrates the profiles of mean flow velocity and mean potential temperature. We recall that all our derivations are relevant to the well-developed turbulence regime where molecular transports are negligible compared to turbulent transports so that turbulent fluxes practically coincide with total fluxes. This is the case in our DNS, except for the narrow near-wall viscous-turbulent flow-transition layers. Data from these layers, obviously irrelevant to the turbulence regime we consider, are shown by grey points in the figures and are ignored in fitting procedures.

In further analysis, we primarily utilise $z/L$ as a stratification parameter instead of Ri or Ri$_f$ because it offers a better dynamic range in our experiments. While Ri remains practically constant in each DNS run and Ri$_f$ is limited in its growth, the parameter $z/L$ is determined by the distance from the walls, thus varying significantly in every DNS run.

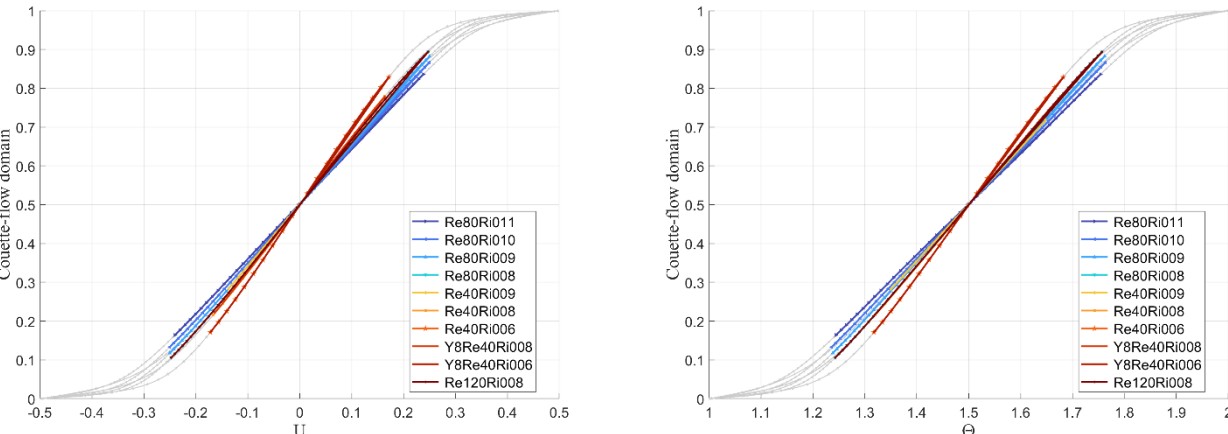

**Figure 1: Profiles of mean flow velocity and mean potential temperature in stably stratified turbulent plane Couette flow. Light grey dots belong to the viscous sublayer.**

Numerical simulation of stably stratified turbulent Couette flow was performed using the unified DNS-, LES- and RANS-code developed at the Moscow State University (MSU) and the Institute of Numerical Mathematics (INM) of the Russian Academy of Science (see, Mortikov, 2016; Mortikov et al., 2019; Bhattacharjee et al., 2022; Debolskiy et al., 2023; Gladskikh et al., 2023, Zasko et al., 2023). The code is designed for high-resolution simulations on modern-day HPC systems. The DNS part of the code solves the finite-difference approximation of the incompressible Navier-Stokes system of equations under the Boussinesq approximation. Conservative schemes on the staggered grid (Morinishi et al., 1998; Vasilyev, 2000) of 4th-order accuracy are used in horizontal direction, while in the vertical direction the spatial approximation is restricted to 2nd-order accuracy with near-wall grid resolution refinement sufficient to resolve near-wall viscous region. The time step used in the simulations was determined by Courant–Friedrichs–Lewy (CFL) restrictions, with CFL maintained at approximately 0.1 in all runs. This corresponds to a value of $u_*^2 \Delta t/\nu$ on the order of 0.01. The projection method (Brown et al., 2001) is used for the time-advancement of momentum equations coupled with the incompressibility condition, while the multigrid method is applied to solve the Poisson equation to ensure that the velocity is divergence-free at each time step. For the Couette flow periodic boundary conditions are used in the horizontal directions, and no-slip/no-penetration conditions are set on the channel walls for the velocity. The stable stratification is maintained by prescribed Dirichlet boundary conditions on the potential temperature. In all experiments, the value of molecular Prandtl number (ratio of kinematic viscosity and thermal diffusivity of the fluid) was fixed at 0.7 based on its typical value for air (Monin and Yaglom, 1971). The simulations were performed for a wide range of Reynolds numbers, Re, defined by the wall velocity difference, channel height and kinematic viscosity: from

40000 up to 120 000 (see Table 1). All experiments were carried out using the resources of MSU and CSC HPC centers. For the maximum Re values achieved the numerical grid consisted of more than $2 \times 10^8$ cells and the calculations used about 10 000 CPU cores.

**Table 1: Overview of DNS experiments and key parameters.**

| DNS run name | Re $(UH/\nu)$ | $Ri_b$ $(\beta\Theta/U^2)$ | Grid size | Domain $(H)$ | $Re_\tau$ $(u_* H/\nu)$ | Viscous sublayer $(z < 50\nu/\tau^{1/2})$ | CPU runtime $(s)$ | Averaging time $(Tu_*/H)$ |
|---|---|---|---|---|---|---|---|---|
| **Re40Ri006** | 40000 | 0.06 | 388 × 260 × 260 | 6 × 4 × 1 | 639.96 | 34.3% | 182180 | 38.40 |
| **Re40Ri008** | 40000 | 0.08 | 388 × 260 × 260 | 6 × 4 × 1 | 525.51 | 43.2% | 165851 | 31.53 |
| **Re40Ri009** | 40000 | 0.09 | 388 × 260 × 260 | 6 × 4 × 1 | 439.96 | 56.5% | 152307 | 26.40 |
| **Y8Re40Ri006** | 40000 | 0.06 | 388 × 516 × 260 | 6 × 8 × 1 | 639.30 | 34.3% | 316204 | 38.36 |
| **Y8Re40Ri008** | 40000 | 0.08 | 388 × 516 × 260 | 6 x 8 x 1 | 524.21 | 44.2% | 302063 | 31.45 |
| **Re80Ri008** | 80000 | 0.08 | 772 × 516 × 516 | 6 × 4 × 1 | 1001.11 | 21.2% | 891598 | 30.03 |
| **Re80Ri009** | 80000 | 0.09 | 772 × 516 × 516 | 6 × 4 × 1 | 912.07 | 23.5% | 946772 | 27.36 |
| **Re80Ri010** | 80000 | 0.10 | 772 × 516 × 516 | 6 × 4 × 1 | 816.91 | 26.7% | 936989 | 24.51 |
| **Re80Ri011** | 80000 | 0.11 | 772 × 516 × 516 | 6 × 4 × 1 | 684.19 | 32.8% | 961394 | 20.53 |
| **Re120Ri008** | 120000 | 0.08 | 772 × 516 × 516 | 6 × 4 × 1 | 1328.72 | 21.2% | 848043 | 26.57 |

For each Reynolds number, we conducted a series of experiments. Beginning with neutral conditions (no imposed gradient of the mean potential temperature), we incrementally increased the bulk Richardson number, which characterises the stable

stratification, in each successive experiment. By gradually increasing stability in each experiment, we were able to cover a wide range of Ri values, extending from neutral to stably stratified states. In each run, the turbulent flow was allowed sufficient time to develop and reach statistical steady-state conditions, which required a spin-up period of at least 15 $H/u_*$ periods. This ensured that parameters such as the total momentum flux remained constant and the TKE balance was in a steady state. The fully-developed steady state was used as initial conditions for the higher Ri or Re experiment setups. Additionally, all terms in

the second-order moments budget equations (Eqs. 1-3) were evaluated consistently using the finite-difference approximation used, resulting in negligible residual. This approach enabled us to comprehensively study the characteristics of shear-produced stably stratified turbulence, explicitly resolving all dissipation time scales of turbulent second-order moments.

## 4 Modified EFB closure model for the steady-state regime of turbulence

In the steady-state, Eq. (1) for the vertical component of the turbulent flux of momentum, $\tau$, becomes

$$0 = -2E_z \frac{\partial U}{\partial z} - [\varepsilon_\tau - \beta F_x - Q_{13}].$$ (22)

Following Zilitinkevich et al. (2007, 2013) we define the sum of all terms in square brackets on the r.h.s. of Eq. (22) as the "effective dissipation":

$$\varepsilon_\tau^{(eff)} = \varepsilon_\tau - \beta F_x - Q_{13} \equiv \frac{\tau}{t_\tau}.$$ (23)

Thus, Eq. (22) becomes

$$0 = -2E_z \frac{\partial U}{\partial z} - \frac{\tau}{t_\tau},$$ (24)

yielding the well-known down-gradient formulation of the vertical turbulent flux of momentum:

$$\tau = -K_M \frac{\partial U}{\partial z}, \quad K_M = 2A_z E_K t_\tau,$$ (25)

where $A_z \equiv E_z/E_K$ is the vertical share of TKE (the vertical anisotropy parameter). Substituting Eq. (25) into Eq. (18), we obtain

$$\left(\frac{\tau}{E_K}\right)^2 = \frac{2A_z}{1 - \mathrm{Ri}_f} \frac{t_\tau}{t_K}.$$ (26)

In Eq. (26) all the variables are exactly resolved numerically in DNS making a detailed investigation on $t_\tau/t_K$ possible. Fig. 2 demonstrates that the dissipation time scale ratio $t_\tau/t_K$ is a function of the stratification parameter $z/L$ rather than a constant. We propose to approximate this function with a ratio of two first-order polynomials:

$$\frac{t_\tau}{t_K} = \frac{C_1^{\tau K} z/L + C_2^{\tau K}}{z/L + C_3^{\tau K}}.$$ (27)

Here, the dimensionless empirical constants are obtained from the best fit of Eq. (27) to DNS bin-averaged data: $C_1^{\tau K} = 0.08$, $C_2^{\tau K} = 0.4$, $C_3^{\tau K} = 2$. The fitting is done using the rational regression model of Curve Fitting Toolbox version: 3.5.13 (R2021a). The ratio of two first-order polynomials is chosen as a simpler fitting function that could provide monotonicity, reasonable smoothness, and clear asymptotes The only three adjustable parameters of this approximation correspond to the function value at $z/L = 0$, the $z/L \rightarrow \infty$ limit, and the transition between them.

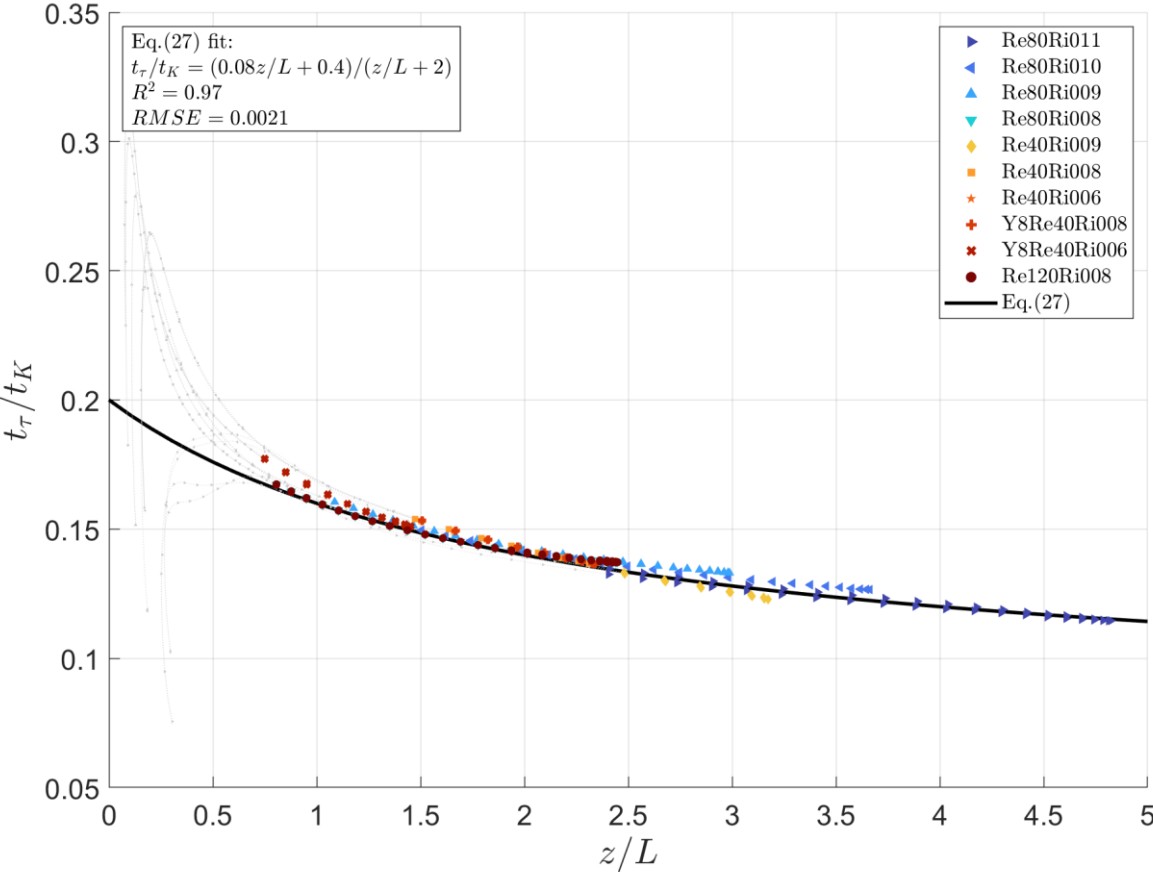

**Figure 2: The ratio of the effective dissipation time scale of $\tau$ and the dissipation time scale of TKE, $t_\tau/t_K$, versus $z/L$. The data used for the calibration are obtained in DNS experiments employing the MSU/INM unified code. Only every 6th data point is presented to increase visibility. For the full dataset, please see Kadantsev and Mortikov, 2024. The near-surface layer essentially affected by molecular viscosity ($0 < z < 50\nu/\tau^{1/2}$) is excluded from the analysis. This sub-layer is represented by the light grey dotted lines. The black solid line shows Eq. (27) with empirical constants $C_1^{\tau K} = 0.08$, $C_2^{\tau K} = 0.4$ and $C_3^{\tau K} = 2$, obtained from the best fit of Eq. (27) to DNS data in the turbulent layer: $z > 50\nu/\tau^{1/2}$ .**

Proceeding to the vertical flux of potential temperature, $F_z$, we derive its steady-state budget equation from Eq. (2):

$$\frac{\partial}{\partial z}\Phi_z^{(F)} = \beta\langle\theta^2\rangle - \frac{1}{\rho_0}\langle\theta\frac{\partial p}{\partial z}\rangle - 2E_z\frac{\partial\Theta}{\partial z} - \varepsilon_F. \tag{28}$$

DNS modelling has shown that $\frac{\partial}{\partial z}\Phi_z^{(F)}$ term to be of the same order of magnitude as $\varepsilon_F$, and it is of the same sign, so we introduce the 'effective dissipation rate' $\varepsilon_F^{(eff)}$:

$$\varepsilon_F^{(eff)} = \varepsilon_F + \frac{\partial}{\partial z}\Phi_z^{(F)} \equiv \frac{F_z}{t_F}. \tag{29}$$

Consequently, Eq. (28) reduces to

$$0 = \beta\langle\theta^2\rangle - \frac{1}{\rho_0}\langle\theta\frac{\partial p}{\partial z}\rangle - 2E_z\frac{\partial\Theta}{\partial z} - \frac{F_z}{t_F}. \tag{30}$$

Traditionally, the pressure term was either assumed to be negligible or declared to be proportional to $\beta\langle\theta^2\rangle$ term (see Zilitinkevich et al. 2007; 2013). However, our DNS data have shown that it is neither negligible nor proportional to any other term in the budget equation, Eq. (30). Instead, we found it is well approximated by a linear combination of the production and transport terms of Eq. (30) (see Fig. 3):

$$\frac{1}{\rho_0}\langle\theta\frac{\partial p}{\partial z}\rangle = C_\theta\beta\langle\theta^2\rangle + C_\nabla 2E_z\frac{\partial\Theta}{\partial z}. \tag{31}$$

The dimensionless constants $C_\theta = 0.82$ and $C_\nabla = -0.80$ are obtained from the best fit of Eq. (31) to DNS data.

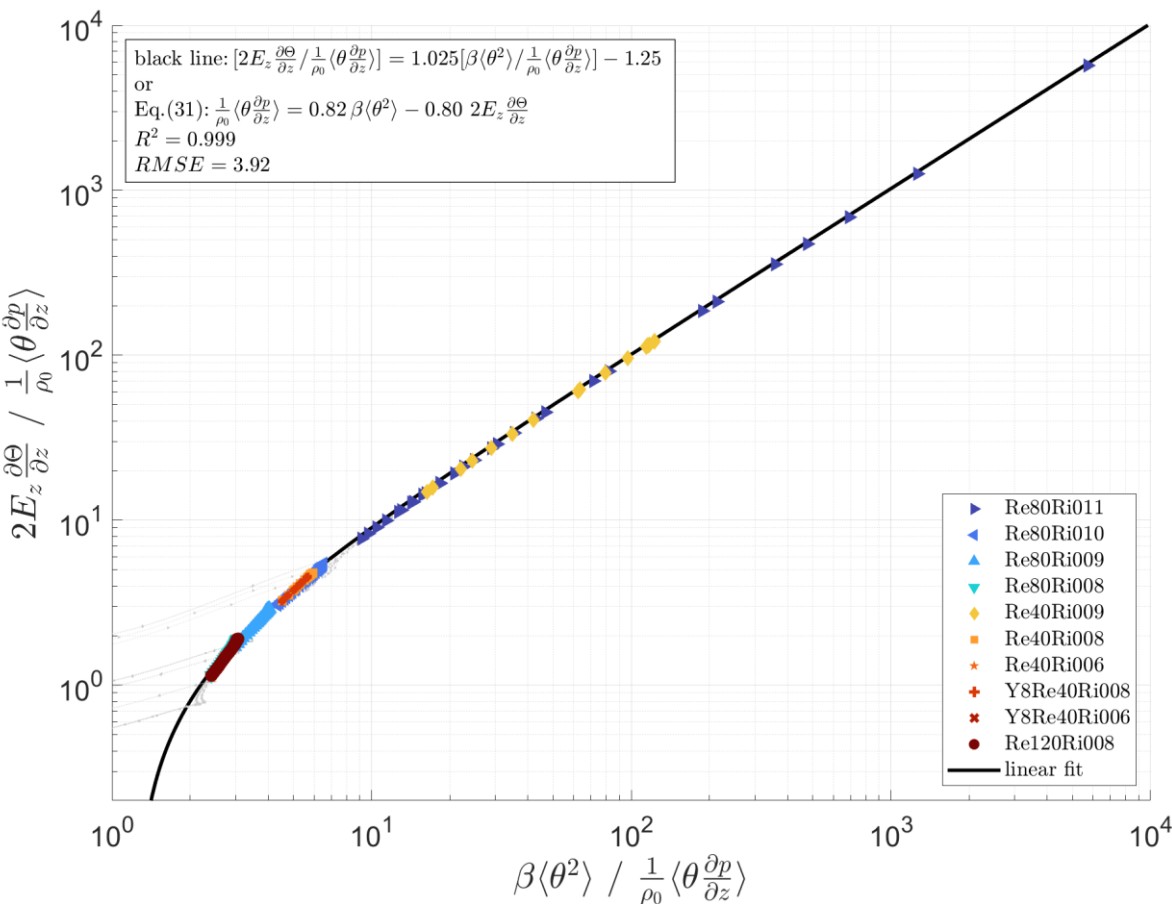

**Figure 3: Comparison of two terms, $\beta\langle\theta^2\rangle/\frac{1}{\rho_0}\langle\theta\frac{\partial p}{\partial z}\rangle$ and $2E_z\frac{\partial\Theta}{\partial z}/\frac{1}{\rho_0}\langle\theta\frac{\partial p}{\partial z}\rangle$, after the same DNS for stably stratified Couette flow. The black solid line represents the linear dependency of the latter on the former, which turns into Eq. (31) after multiplication by $\frac{1}{\rho_0}\langle\theta\frac{\partial p}{\partial z}\rangle$ and simple recombination. The fitting coefficients are $C_\theta = 0.82$ and $C_\nabla = -0.80$.**

Substituting Eq. (31) into Eq. (30), we rewrite the budget equation as

$$0 = (1 - C_\theta)\beta\langle\theta^2\rangle - (1 + C_\nabla)2E_z\frac{\partial\Theta}{\partial z} - \frac{F_z}{t_F}. \tag{32}$$

Substituting Eq. (15) for $\langle\theta^2\rangle$ into Eq. (32) allows expressing $F_z$ through familiar temperature-gradient expression:

$$F_z = -K_H\frac{\partial\Theta}{\partial z}, \quad K_H = \left[(1 + C_\nabla) - (1 - C_\theta)\frac{E_P}{A_zE_K}\right]2A_zE_Kt_F. \tag{33}$$

Substituting Eq. (33) into Eq. (14), gives

$$\frac{F_z^2}{E_\theta E_K} = 2\left[(1 + C_\nabla)A_z - (1 - C_\theta)\frac{E_P}{E_K}\right]\frac{t_F}{t_\theta}. \tag{34}$$

Next, the turbulent Prandtl number, defined as $\mathrm{Pr}_T = K_M/K_H$, is given by

$$\mathrm{Pr}_T = \frac{t_\tau}{t_F}\Big/\left[(1 + C_\nabla) - (1 - C_\theta)\frac{E_P}{A_zE_K}\right]. \tag{35}$$

Eqs. (34) and (35) provide us with two constrains on the function in the square brackets. First, the left-hand side of Eq. (34) is non-negative by definition, implying the same requirement for the right-hand side of the equation. Second, the turbulent Prandtl number grows with increase of the gradient Richardson number, $\mathrm{Pr}_T|_{(z/L\to\infty)} \to Ri/R_\infty$, requiring the function in the square 270 brackets to approach zero under extreme stratification. This leads us to the next approximation (see Fig. 4):

$$\frac{1 - C_\theta}{1 + C_\nabla}\frac{E_P}{A_zE_K} = 1 - e^{-C_{Pr}z/L}. \tag{36}$$

This function monotonically decreases from 1 to 0 as $0 < z/L < \infty$, satisfying our requirements with $C_{Pr} = 0.65$. The observed spread of data points might be explained by the simulation time being insufficient to reach a fully statistical steady state for this specific ratio. Although the fully developed steady state was achieved (verified using the standard criterion of 275 stabilized TKE, which showed no significant fluctuations over time), the parameters involving ratios of temperature fluctuations $\theta$ might require additional time to stabilize. We believe that increasing the experiment time would decrease the spread, but we leave the validation of this hypothesis for future studies.

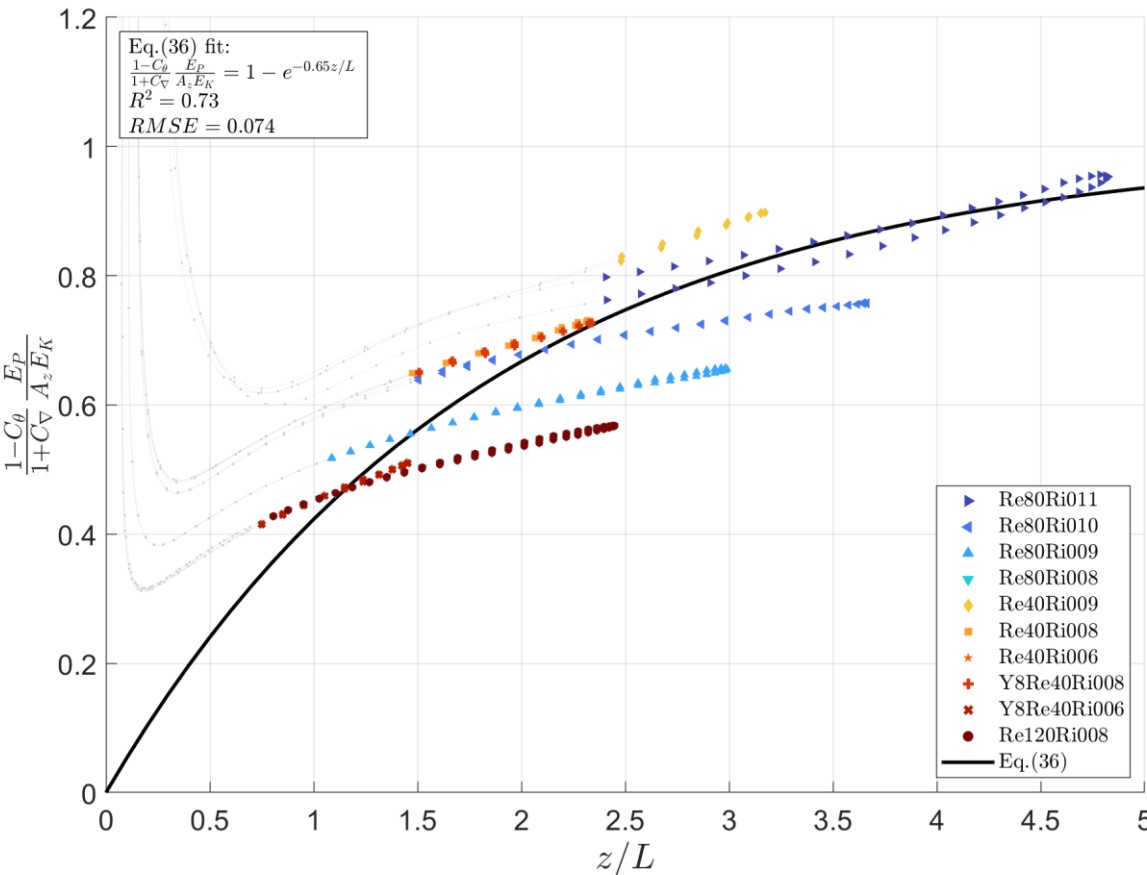

**Figure 4: The ratio of two terms from the square brackets of Eq. (34) versus $z/L$. Same data as in Fig. 2. The black solid line shows Eq. (36) with empirical constant $C_{Pr} = 0.65$, obtained from the best fit of Eq. (34) to DNS data in the turbulent layer: $z > 50\nu/\tau^{1/2}$.**

It leads us to a similar approximation of $t_\tau/t_F$ (see Fig. 5):

$$\frac{t_\tau}{t_F} = \Pr_T(1 + C_\nabla)\left[1 - \frac{1-C_\theta}{1+C_\nabla}\frac{E_P}{A_z E_K}\right] = C_1^{\tau F} e^{-C_2^{\tau F} z/L}. \tag{37}$$

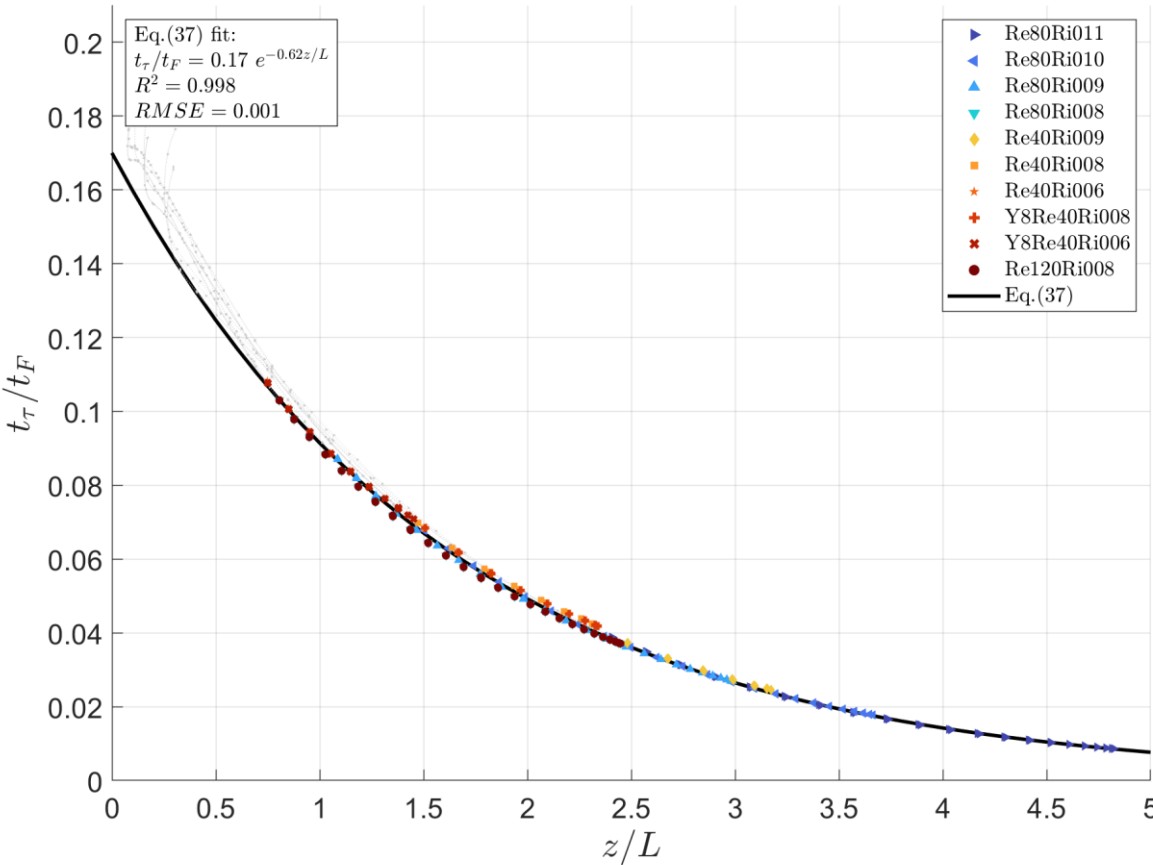

**Figure 5: The ratio of the effective dissipation time scales of $\tau$ and $F_z$, $t_\tau/t_F$, versus $z/L$. Same data as in Fig. 2. The black solid line shows Eq. (37) with empirical constants $C_1^{\tau F} = 0.17$ and $C_2^{\tau F} = 0.62$, obtained from the best fit of Eq. (37) to DNS data in the turbulent layer: $z > 50\nu/\tau^{1/2}$.**

Now, to complete the closure, we need to determine one more dimensionless ratio, $t_\theta/t_K$. It is explicitly required for the ratio of turbulent energies, $E_P/E_K$, and consequently for $A_z$ through Eqs. (20) and (36). We approximate it once again with the ratio of two first-order polynomials:

$$\frac{t_\theta}{t_K} = \frac{C_1^{\theta K} z/L + C_2^{\theta K}}{z/L + C_3^{\theta K}}. \tag{38}$$

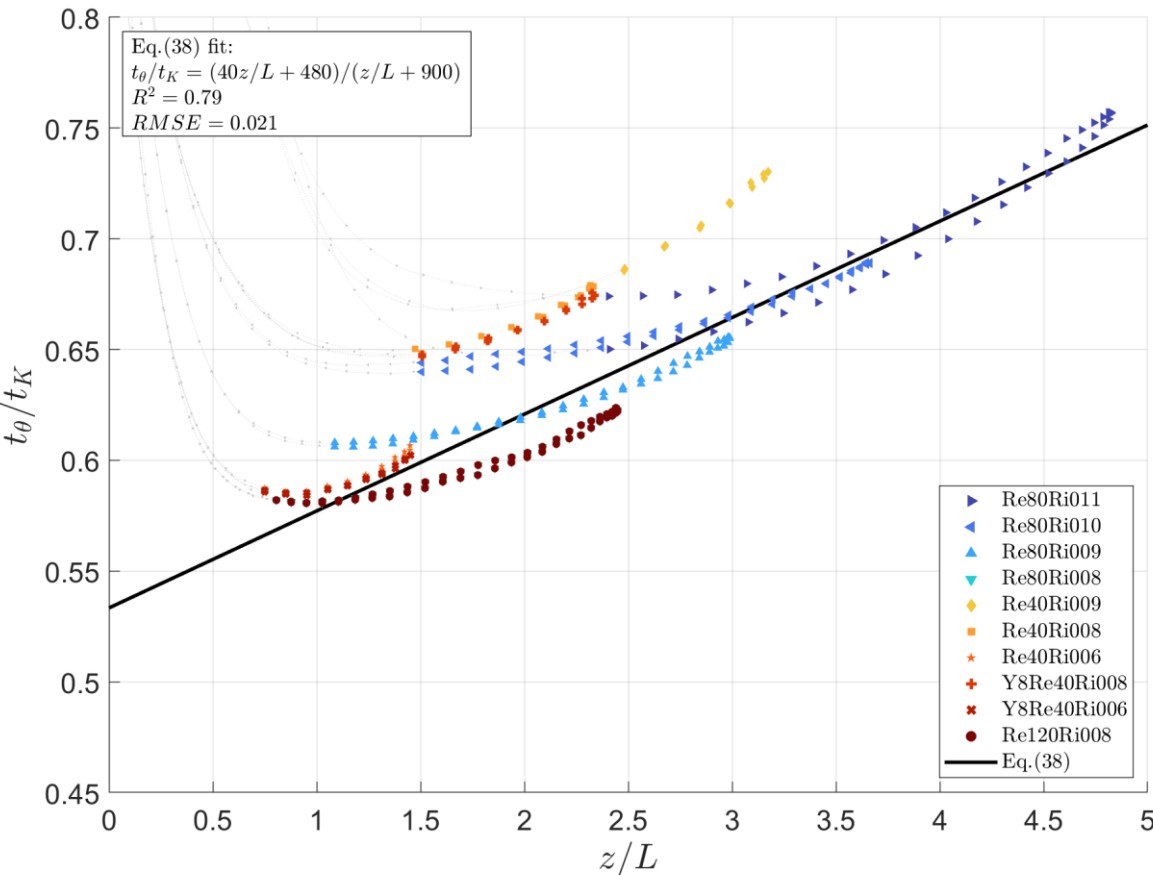

**Figure 6:** The ratio of the dissipation time scale of $\langle \theta^2 \rangle$ and the dissipation time scale of TKE, $t_\theta/t_K$, versus $z/L$. Same data as in Fig. 2. The black solid line shows Eq. (38) with empirical constants $C_1^{\theta K} = 40$, $C_1^{\theta K} = 480$ and $C_1^{\theta K} = 900$, obtained from the best fit of Eq. (38) to DNS data in the turbulent layer: $z > 50\nu/\tau^{1/2}$.

With Eq. (38), our turbulence closure is now complete, allowing us to proceed with the verification process using quantities not utilized in the fitting procedures. Fig. 7 provides empirical evidence supporting the stability dependencies given by Eqs. (20, 26, 27, 34-38). Table 2 summarises the proposed approximations and provides a summary of the resulting turbulent closure.

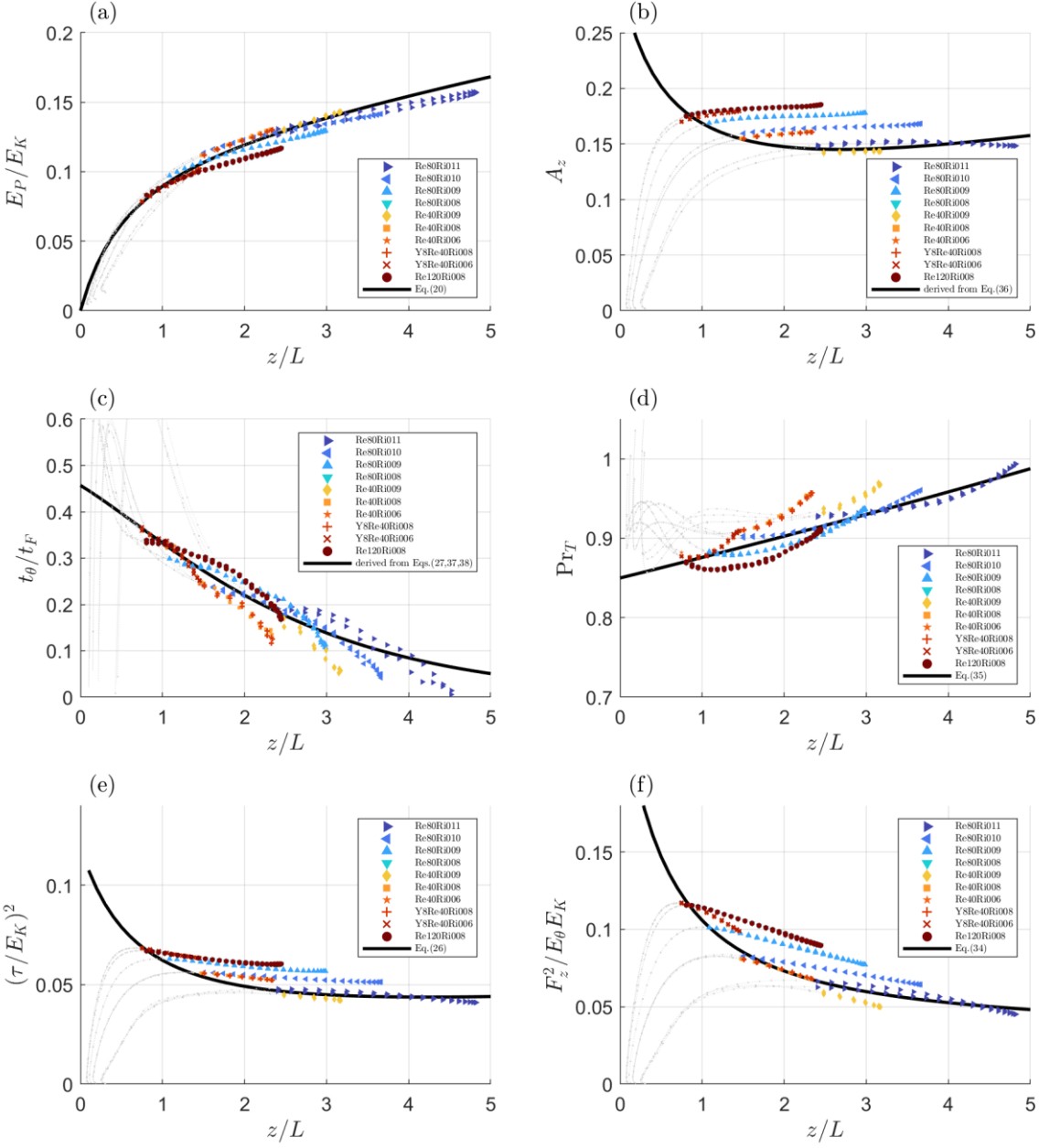

**Figure 7: Validating the closure with quantities not utilized in the fitting procedures. Panel (a) shows the TPE to TKE ratio, $E_P/E_K$; panel (b) shows the vertical share of TKE, $A_z$; panel (c) demonstrates the ratio of dissipation time scales of $\langle\theta^2\rangle$ and $F_z$; panel (d) shows the turbulent Prandtl number, $\mathrm{Pr}_T$; panel (e) shows the squared dimensionless turbulent flux of momentum, $(\tau/E_K)^2$; and panel (f) shows the squared dimensionless turbulent flux of potential temperature, $F_z{}^2/E_\theta E_K$. All quantities are plotted against $z/L$.**

**The black solid lines correspond to theoretical predictions demonstrating acceptable-to-great agreement with the DNS data in the turbulent layer: $z > 50\nu/\tau^{1/2}$. Empirical data are from the same sources as in Fig. 2. No fitting has been performed for this figure.**

For practical reasons, most operational numerical weather prediction and climate models parameterize these dimensionless ratios as functions of the gradient Richardson number rather than $z/L$. This preference arises from the fact that the gradient Richardson number is defined solely by mean quantities, namely buoyancy and shear productions, which in practice imposes fewer computational restrictions on the model's time step. Since $\text{Ri} = \text{Pr}_T \text{Ri}_f$ and both $\text{Pr}_T$ and $\text{Ri}_f$ are defined as functions
of $z/L$ by Eqs. (35) and (21), respectively, we can derive an expression for the gradient Richardson number Ri as the function of $z/L$, shown in Fig. 8:

$$\text{Ri} = \text{Ri}_f \frac{C_1^{\tau F}}{1+C_{\overline{V}}} e^{-(C_{Pr}-C_2^{\tau F})z/L}. \tag{39}$$

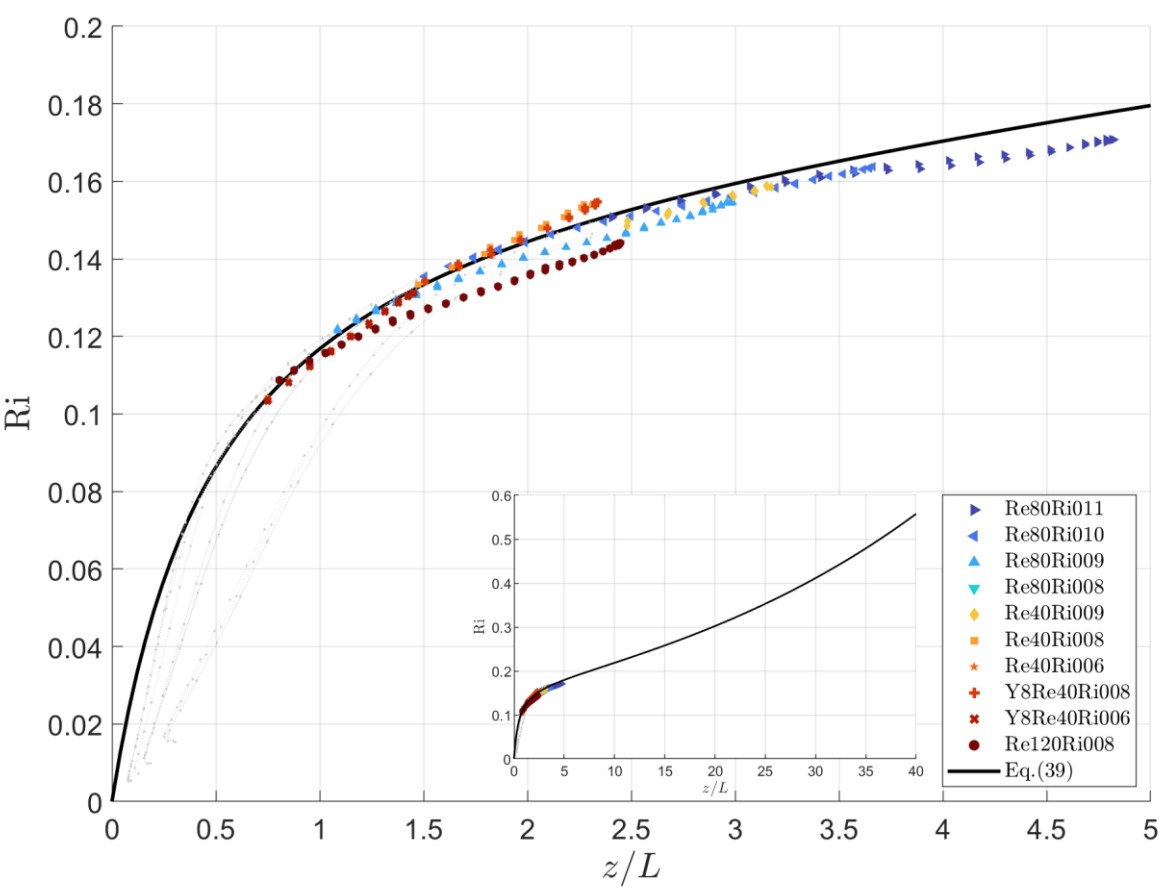

**Figure 8: Resulting approximation of the gradient Richardson number, Ri, after Eq. (39). compared to the exact solution (panel a)**
315 **and relative error of this approximation as a function of gradient Richardson number, Ri (panel b). The black solid line corresponds to theoretical derivation, that shows good agreement with the DNS data in the turbulent layer: $z > 50\nu/\tau^{1/2}$. Empirical data are from the same sources as in Fig. 2. No fitting has been performed for this figure.**

**Table 2: Proposed approximations and resulting revised turbulent parameters of EFB closure.**

| Variable | Approximation / theoretical derivation | Empirical constants | $R^2$ | RMSE | Equation number |
|---|---|---|---|---|---|
| $\dfrac{t_\tau}{t_K}$ | $\dfrac{C_1^{\tau K} z/L + C_2^{\tau K}}{z/L + C_3^{\tau K}}$ | $C_1^{\tau K} = 0.08,\ C_2^{\tau K} = 0.4,\ C_3^{\tau K} = 2$ | 0.97 | 0.0021 | (27) |
| $\dfrac{1}{\rho_0}\langle\theta\dfrac{\partial p}{\partial z}\rangle$ | $C_\theta \beta \langle\theta^2\rangle + C_\nabla 2 E_z \dfrac{\partial \Theta}{\partial z}$ | $C_\theta = 0.82,\ C_\nabla = -0.80$ | 0.999 | 3.92 | (31) |
| $\dfrac{1 - C_\theta}{1 + C_\nabla}\dfrac{E_P}{A_z E_K}$ | $1 - e^{-C_{Pr} z/L}$ | $C_{Pr} = 0.65$ | 0.73 | 0.074 | (36) |
| $\dfrac{t_\tau}{t_F}$ | $C_1^{\tau F} e^{-C_2^{\tau F} z/L}$ | $C_1^{\tau F} = 0.17,\ C_2^{\tau F} = 0.62$ | 0.998 | 0.001 | (37) |
| $\dfrac{t_\theta}{t_K}$ | $\dfrac{C_1^{\theta K} z/L + C_2^{\theta K}}{z/L + C_3^{\theta K}}$ | $C_1^{\theta K} = 40,\ C_2^{\theta K} = 480,\ C_3^{\theta K} = 900$ | 0.79 | 0.021 | (38) |
| $\dfrac{E_P}{E_K}$ | $\dfrac{\mathrm{Ri}_f}{1 - \mathrm{Ri}_f}\dfrac{t_\theta}{t_K}$ | no additional fitting | 0.90 | 0.006 | (20) |
| $A_z$ | $\dfrac{1 - C_\theta}{1 + C_\nabla}\dfrac{E_P}{E_K}\dfrac{1}{1 - e^{-C_{Pr} z/L}}$ | no additional fitting | 0.17 | 0.024 | derived form (36) |
| $\dfrac{t_\theta}{t_F}$ | $\dfrac{t_\tau}{t_F}\dfrac{t_\theta}{t_K}\Big/\dfrac{t_\tau}{t_K}$ | no additional fitting | 0.89 | 0.27 | derived from (27, 37, 38) |
| $\mathrm{Pr}_T$ | $\dfrac{t_\tau}{t_F}\dfrac{1}{(1 + C_\nabla) - (1 - C_\theta)\dfrac{E_P}{A_z E_K}}$ | no additional fitting | 0.76 | 0.017 | (35) |
| $\left(\dfrac{\tau}{E_K}\right)^2$ | $\dfrac{2 A_z}{1 - \mathrm{Ri}_f}\dfrac{t_\tau}{t_K}$ | no additional fitting | 0.61 | 0.008 | (26) |
| $\dfrac{F_z^2}{E_\theta E_K}$ | $2\left[(1 + C_\nabla) A_z - (1 - C_\theta)\dfrac{E_P}{E_K}\right]\dfrac{t_F}{t_\theta}$ | no additional fitting | 0.77 | 0.014 | (34) |
| $\mathrm{Ri}$ | $\mathrm{Ri}_f \dfrac{C_1^{\tau F}}{1 + C_\nabla} e^{-(C_{Pr} - C_2^{\tau F}) z/L}$ | no additional fitting | 0.90 | 0.005 | (39) |

**5 Concluding remarks**

For many years, our understanding of dissipation rates for turbulent second-order moments has been hindered by a lack of direct observations in fully controlled conditions, particularly in a strongly stable stratification. To address this limitation, we

conducted topical DNS experiments of stably stratified Couette flows. The main finding of this study is that the ratios of the dissipation time scales of the basic second-order moments depend on the temperature stratification (e.g., characterised by the gradient Richardson number), contrary to the traditional assumption of them being proportional to a single universal dissipation time scale.

This finding laid the foundation for empirically approximating these ratios with simple universal functions of stability parameters, valid for a wide range of stratifications. Consequently, this allowed us to refine the EFB turbulent closure by accounting for dissipation time scales that are intrinsic to the basic second-order moments. As a result, the revised formulations for eddy viscosity and eddy conductivity reveal greater physical consistency in stratified conditions, thereby enhancing the representation of turbulence in numerical weather prediction and climate modelling.

We have also observed that the dimensionless parameters involving $\theta$ fluctuations demonstrate a wider spread of values within and across the DNS experiments, making it more challenging to approximate them with stability functions. This suggests that the stabilisation time for these parameters may be significantly longer than for TKE components.

It is important to note that our DNS experiments were limited to gradient Richardson numbers up to $Ri = 0.17$. Any data reliably indicating different asymptotic values of the time scale dimensionless ratios or demonstrating their different dependency on the temperature stratification would pose the need for readjusting the proposed parameterization.

We deliberately avoided discussing intermittency issues: for that one needs to determine higher-order two-point (or multi-point) moments. Intermittency is important for small-scale effects, and intermittency implies that higher-order moments of velocity and temperature fields have non-Gaussian statistics. In this study we focused on larger scales determining one-point second-order correlation functions barely touching one-point third-order correlation functions only when it is necessary. However, addressing this topic would be crucial for advancing numerical simulations towards higher stratifications and warrants detailed investigation.

With these considerations in mind, we believe the most challenging step will be to explicitly explore the transitional region between traditional weakly-stratified turbulence and extremely stable stratification, where the behaviour of the turbulent Prandtl number shifts from nearly constant to a linear function with respect to the gradient Richardson number. Investigating this phenomenon would require unprecedented computational resources for DNS or specialised in-situ or laboratory experiments.

**Code and data availability**

The DNS code is available by GitLab at http://tesla.parallel.ru. The datasets generated and analysed during the current study are available at https://doi.org/10.23728/b2share.7a1d875b872748c7bf566ece352c0a10.

## Author contribution

EK conceptualised the paper, performed data analysis, wrote the initial text, and prepared the figures. EM contributed to the conceptualisation of the study, developed the DNS code, and performed the numerical simulations. AG contributed to the conceptualisation of the study and code development. NK and IR contributed to the conceptualisation of the study and assisted with literature overview and manuscript editing.

## Competing interest

The authors declare that they have no conflict of interest.

## Acknowledgements

This paper was not only inspired by but also conducted under the supervision of the esteemed Prof. Sergej Zilitinkevich, who unfortunately is no longer with us. We wish to express our profound gratitude to Sergej for the incredible honour of collaborating with him and for the immense inspiration he generously bestowed upon us.

The authors would like to acknowledge the following funding sources for their support in conducting this research: the project "Research Infrastructures Services Reinforcing Air Quality Monitoring Capacities in European Urban & Industrial Areas" (RI-URBANS, grant no. 101036245) and the Academy of Finland project HEATCOST (grant no. 334798). This work was also partially supported by the FSTP project no. 124042700008-6 "Research in geophysical boundary layers and the development of new modelling approaches for Earth system models" within the program "Improvement of the global world-level Earth system model for research purposes and scenarios forecasting of climate change", RSCF grant no. 21-71-30003 (development of the DNS model) and by MESRF as part of the program of the Moscow Center for Fundamental and Applied Mathematics under agreement no. 075-15-2022-284 (DNS of stably stratified Couette flow). DNS experiments were carried out using the CSC HPC center infrastructure and the shared research facilities of the HPC computing resources at MSU.

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
