# Peer review of "On dissipation time scales of the basic second-order moments: the effect on the Energy and Flux-Budget (EFB) turbulence closure for stably stratified turbulence"

_EGUsphere, 2023_

## Referee Comment (RC2)

**Review of On dissipation time scales......., Kadantsev et al., 2023-3164**

This review references detailed comments to the line(s) in the manuscript.

GENERAL

This paper, like many others in the two centuries that have elapsed since Navier first formulated equations of fluid flow and the one and a half since Stokes corrected them, stumbles on the closure problem and on the concept of dissipation. It is my contention that application of the Langevin equation to the atmosphere results in the conclusion that the emergence of fluid flow and dissipation are intimately linked at the molecular level. References [1,2,3,4] have argued so; "Direct Numerical Simulation" should really entail this approach rather than some arbitrary scale assumption.

COMMENTARY

17-18: In a coupled nonlinear system like the atmosphere, "control" is a slippery concept. Particularly when the real atmosphere's boundaries are far less restrictive than those employed here. Laminar flow of any sort is not to be found in the air, let alone such an artificial system such as Couette flow.

31: Kolmogorov's theory has been reformulated for the atmosphere; see for example [5,6].

35-41: "Closure" betrays the real difficulty. The need for a bottom up, molecular approach has been largely ignored, but was pointed out in [1,2,3].

45-50: See references [7,8,9] for a discussion of *Ri.*

60-79: The debate about dissipation ignores the reality of the Langevin equation and the approach to dissipation that emerges from a bottom up, molecular dynamics approach [4,10,11].

153 et seq: Couette flow has little physical reference to any atmospheric flow. The boundary conditions are far too restrictive. DNS is not direct. Molecular dynamics would qualify and is now almost within reach of current computational performance.

205: "dissipation time scale" is covering some major difficulties. See [4,10,11].

299-316: Dissipation is the process defining an operational temperature; it is infrared radiation to space.

REFERENCES

1. von Neumann, J. L. Recent Theories of Turbulence, in *Collected Works of John von Neumann,* Vol. 6, pp. 437-472, Macmillan, 1963.

2. Grad, H. Principles of the Kinetic Theory of Gases, in *Handbuch der Physik,* Band **12,** S. Flugge, ed., pp. 205-294, Springer Verlag, Berlin, 1958.

3. Tuck, A. F. *ATMOSPHERIC TURBULENCE: A Molecular Dynamics Perspective.* Oxford University Press, Oxford, 2008.

4. Tuck, A. F. Scaling up: molecular to meteorological via symmetry breaking and statistical multifractality. *Meteorology,* **1,** 4-28 (2022)

5. Schertzer, D.; Lovejoy, S. Physical modeling of rain and clouds by anisotropic scaling multiplicative processes. *J. Geophys.Res. D,* **92,** 9693-9714 (1987).

6. Lovejoy, S.; Schertzer, D. *The Weather and Climate: Emergent Laws and Multifractal Cascades.* Cambridge University Press, Cambridge, UK. (2013).

7. Lovejoy, S.; Tuck, A. F.; Hovde, S. J.; Schertzer, D. Is isotropic turbulence relevant in the atmosphere? *Geophys. Res. Lett.* **34,** L15082 (2007).

8. Lovejoy, S.; Tuck, A. F.; Hovde, S. J.; Schertzer, D. Do stable atmospheric layers exist? *Geophys. Res. Lett.* **35,** L01802 (2008).

9. Hovde, S. J.; Tuck, A. F.; Lovejoy, S.; Schertzer, D. Vertical scaling of temperature, wind and humidity fluctuations: dropsondes from 13 km to the surface of the Pacific Ocean. *Int. J. Remote Sensing* **32,** 5891-5918 (2011).

10. Tuck, A. F. Turbulence: vertical shear of the horizontal wind, jet streams, symmetry breaking and Gibbs free energy. *Atmosphere,* **12,** 1414 (2021).

11. Tuck, A. F. Air temperature intermittency and photofragment excitation. *Meteorology,* **2,** 445-463 (2023).

RECOMMENDATION

If the journal wishes to continue the largely unsuccessful grappling with turbulence that has characterized the last two centuries, then publish this paper - it is better than most of the genre. But if so, the authors should acknowledge, however briefly, some of the difficulties outlined above.

---

## Author Comment (AC1)

**'On dissipation time scales …' preprint, response to Anonymous Referee #3**

Review by Anonymous Referee #3
(https://egusphere.copernicus.org/preprints/egusphere-2023-3164#RC3).

*Authors' response in italics, highlighted by light grey colour.*

This manuscript aims to provide additional insight to the effects of stratification on turbulence properties and consequences for how we model those. The results are based on DNS data and show some promising results and valuable discussion, however, there are some odd choices in the analysis that result in more confusion than clarity. Furthermore, the abstract and introduction includes discussion of conditions up to extreme static stability which the results of the paper does not provide results for. At the end of the manuscript (line 310) it is noted that the DNS experiments were limited to gradient Richardson numbers up to Ri=0.2. This statement does not align with the Figures that presents data up to z/L = 5 which is confusing. This discrepancy goes to the heart of my problem with the analysis which is the introduction of performing the analysis using z/L as stability parameter which is just introduced without proper motivation on line 145. At the end of the manuscript, the authors then advocate to go back to a Richardson number (in this case the gradient Ri) with the motivation "for practical reasons". I would like to see the analysis performed using Ri as the stability parameter throughout which I anticipate would provide a more straightforward analysis. Furthermore, I dislike the extrapolation outside of the DNS parameter space, for example the exponential growth far outside the range of the DNS results in Figure 6d. Presenting the results in this way discredits the results. In conclusion, the manuscript needs considerable rewriting before it can be properly judged for publication.

*We appreciate the reviewer's comments. First, we need to apologise for the misleading estimate of maximum presented gradient Richardson number which in fact was only $Ri = 0.12$. This oversight (a typo) has now been corrected.*

*We prefer to show the $z/L$ dependences rather than the $Ri$ dependences in our analysis for a practical reason. In Couette flow $Ri$ barely changes within the fully-developed turbulence layer due to minimal gradient variations (see new Figure 1 in the manuscript), while $z/L$ provides a better dynamic range, given that $L$ remains practically constant while $z$ is determined by the distance from the walls (for details, see Figure 1 in Zilitinkevich et al., 2019):*

[Figure]

[Figure]

*This clarification is now added in the beginning of Section 3.*

Line 25: There is no analysis of extreme static stability presented.

*We have revised the wording to reflect the range of stability conditions studied more accurately.*

Line 144: There is no motivation for using this conversion from flux Ri to z/L (which is a stability parameter and not a stratification parameter). I understand from reading the Acknowledgements that Prof Zilintikevich was instrumental for the project, maybe this remains as one of his ideas. However, if it does not make sense for the continued analysis, it should be removed. That would be in the spirit of Sergej, whom I knew and also worked with.

*The clarification of stratification parameter preference is now added in the beginning of Section 3.*

Line 152: It is not correct to write "empirical validation". First DNS data is not empirical data, and second, the data is used for evaluation not validation.

*The title of Section 3 has been changed to 'Methods' to avoid confusion. Following the recommendation of the Referee we have replaced 'empirical' by a more accurate term 'obtained from DNS experiment'.*

Line 171: More details on how the prescribed Dirichlet boundary is imposed to maintain the stable stratification is needed.

*We have added clarifications on the DNS setup in the paper. The stable stratification is maintained by prescribed Dirichlet boundary conditions on the potential temperature. This, along with prescribed Dirichlet boundary conditions on the velocity field, allows us to fix the Reynolds number (based on the wall velocity difference and channel height) and the bulk Richardson number (based on the wall velocity and temperature differences and channel height) in each experiment.  It is important to note that in this case, the friction velocity and the*

*potential temperature flux (as well as the Obukhov length scale) are computed during the model run, rather than being prescribed.*

Line 172: Why did you chose to fix a value of the molecular Pr number to 0.7. What is that based on?

*We chose to fix the value of the molecular Prandtl number to 0.7 based on its typical value for air. The clarification has been added.*

Line 183: When reaching the end of this description, there are still missing information on how the DNS experiments were conducted. How many simulations? Initial conditions? Time step? At what stratification? When did the simulation reached statistical steady-state, to what accuracy? Again, it is stated "turbulence up to extreme static stability" which is not that case. How do you know that you are resolving all dissipation time scales? Do you have any numerical diffusion? The experimental parameters could be summarized in a Table. It would also add to the manuscript if the various experiments were color-coded in the Figures so they can be identified.

*We have made several updates to the manuscript. A table summarising DNS parameters has been added, covering ten different experiments presenting well-developed turbulence. Additionally, we have included a more comprehensive description of the DNS methodology. The figures have been replotted to enable the identification of different experiments.*

Line 187: Is it correct to interpret this statement as buoyancy is a dissipative term in stratified conditions?

*Correct. Indeed, the molecular-viscosity dissipation term is relatively small, with the dissipative role being largely fulfilled by the pressure-shear correlations and the horizontal turbulent transport of potential temperature (see figure below).*

[Figure]

Line 197: It is really not clear to me why you choose to plot the results as function of z/L when you have Ri_f in the equation. Furthermore, I think it would be good to remove the near-neutral DNS results as they are not credible anyway in all figures, that would lead to improved visibility in the various panels. The fitted line in Figure 1 cross z/L = 0 at the value 0.2, is that a given? Do you have neutral DNS to constrain that?

*The clarification of the stability parameter preference is now added at the beginning of Section 3. The viscous sublayer points have been toned down in all figures to enhance visibility. The ratio of the effective dissipation time scale of τ to the dissipation time scale of TKE was found to be 0.2 at z/L = 0 as a result of fitting DNS data of stably stratified Couette flow; this should be considered an extrapolation. While performing DNS for neutrally stratified flow would confirm or correct this value, we will leave this for future studies as this work is focused on stably stratified turbulence.*

Line 199: Why to you propose a ration of two first-order polynomials? That is a quite advanced fitting, did you try simpler representation of is the proposition based on any theoretical argument?

*The ratio of two first-order polynomials is chosen as a simpler fitting function that could provide monotonicity, reasonable smoothness, and clear finite asymptotes. All three adjustable parameters of this approximation are easy to*

*understand: the function value at $z/L = 0$, the $z/L \rightarrow \infty$ limit, and the transition between them. The clarification is now added to the manuscript.*

Figure 2: The labels are very unclearly written, or unnecessary complicated. I assume you are dividing with the whole left part of Eq 31 but the label it is not clear.

*Correct. The readability of labels in the Figure are now improved.*

Line 252: Would be good with some references here for this discussion, there are empirical results for how asymmetry varies.

Figure 5: Could be interesting to see how this would fair with other assumptions for Az. The DNS results are quite variable.

*After lengthy discussion, we decided to approximate $A_z$ as a function of z/L, in line with other dimensionless parameters, to maintain consistency in our methodological approach, without altering the essence of the paper. Additionally, references to existing approximations of $A_z$ were included.*

Line 281-285: See discussion above regarding the stability parameter, the discussion here is not very insightful.

*We believe that the revised explanation of stability parameter preference makes this part more insightful.*

Line 286: Why is the function a polynomial of the 5th order?

*Since $Ri = \Pr_T Ri_f$ , one might substitute Eqs. (20, 21, 27, 35, 40) into Eq. (36) and perform arithmetic operations, resulting in a ratio of two 5th-degree polynomials. This implies that obtaining z/L after knowing $Ri$ would require solving a polynomial equation of the 5th degree.*

*However, with the recent changes to the approximations in the revised manuscript, this approach is no longer valid, and the approximation for $Ri_f$ vs $Ri$ is required.*

Line 310: I do not understand why you show results that are outside of what the DNS results support. Overall, the figures need to be of better quality.

*The concluding remarks were clarified. The Figures were redrawn for better quality.*

---

## Author Comment (AC2)

**'On dissipation time scales …' preprint, response to Anonymous Referee #1**

Review by Anonymous Referee #1 (https://doi.org/10.5194/egusphere-2023-3164-RC1).

*Authors' response in italics, highlighted by light grey colour.*

The manuscript is well written, logically structured, and clearly present the novel results. I have only a few rather minor comments to the study and its presentation.

*We would like to thank the reviewer for their interest in our paper and for the valuable comments.*

Introduction in general. (1) Could you indicate clearly the research questions of your study; and (2) briefly present the structure of your manuscript?

*The introduction has been rewritten to emphasise the main research problems of our study and outline the structure of the manuscript.*

Lines 75-80. There is rather rough transition to DNS results, please rewrite the text.

*This part of the paper has been rewritten and the transition to the DNS results has been added.*

Line 89. Wind is a projection of velocity on the horizontal plain, there is no "wind velocity", please correct.

*The notion "wind velocity" has been corrected and replaced by "mean flow velocity".*

Line 89. Why \Theta is bold, is it vector?

*\Theta is scalar, this typo has been now corrected.*

Section 3. Line 175. The paper would benefit from the Table with summary of all DNS experiments and their parameters.

It is also useful to have a Figure with the mean profiles of some DNS runs.

*Table 1 summarising the DNS experiments parameters has been added as well as Figure 1 demonstrating Couette flow mean profiles.*

Line 202. What is "a rational regression model"? What software was used to obtain the regression? This is non-linear regression, what is the method to fit the coefficients? What are confidence intervals for the coefficients?

*We have added the reference and the details of fitting procedure summarised in Table 2.*

Figure 3. There are much more gray dots than red dots, why? Does the scatter in the gray dots mean that thickness of the transitional sub-layer was different in different runs. It would be useful to have a look on a few DNS run results.

*The thickness of the viscous sublayer does indeed vary, as it depends on prescribed viscosity (Re) and stratification (Ri): $0<z<50\nu/\tau^{\wedge}(1/2)$. In this study, we deliberately chose to shade the viscous sublayer in DNS results, focusing on the fully developed turbulence because the EFB closure was designed for the fully developed turbulence rather than the viscous sublayer. The thickness of the viscous sublayer for each experiment has been added to Table 1.*

Figure 4,5,6. What are statistical significance of the presented regressions. Scatter is rather large there, what is R-square (explained part) for these approximations?

*The statistical significance of all approximations has been added to the figures. We have added Table 2 indicating the R-square and RMSE of the approximations.*

Conclusions. It would be helpful to have a brief summary of the obtained closure with all values of coefficients summarized in a Table.

*The brief summary of the closure has been added to Table 2.*

Code and Data. It is reasonable to make available all data used to plot the figures as well as the mean characteristics of DNS runs and mean profiles through a data sharing facility, e.g., ZENODO or similar.

*All data used in the manuscript together with meta-data of DNS runs will be uploaded to b2share.eudat.eu. Please check the revised manuscript for the permanent link.*

---

## Author Comment (AC3)

**'On dissipation time scales …' preprint, response to Anonymous Referee #2**

Review by Anonymous Referee #2 (https://editor.copernicus.org/index.php?_mdl=msover_md&_jrl=778&_lcm=oc108lcm109w&_acm=get_comm_sup_file&_ms=117323&c=260754&salt=13916850601854574235).

*Authors' response in italics, highlighted by light grey colour.*

This review references detailed comments to the line(s) in the manuscript.

**GENERAL**

This paper, like many others in the two centuries that have elapsed since Navier first formulated equations of fluid flow and the one and a half since Stokes corrected them, stumbles on the closure problem and on the concept of dissipation. It is my contention that application of the Langevin equation to the atmosphere results in the conclusion that the emergence of fluid flow and dissipation are intimately linked at the molecular level. References [1,2,3,4] have argued so; "Direct Numerical Simulation" should really entail this approach rather than some arbitrary scale assumption.

*We would like to thank the reviewer for the interest in our paper and for the valuable comments.*

**COMMENTARY**

17-18: In a coupled nonlinear system like the atmosphere, "control" is a slippery concept. Particularly when the real atmosphere's boundaries are far less restrictive than those employed here. Laminar flow of any sort is not to be found in the air, let alone such an artificial system such as Couette flow.

*We performed a series of DNS of stably stratified turbulent plane Couette flow for a wide range of Reynolds numbers defined by the wall velocity difference, channel height, and kinematic viscosity, up to Re = 120 000. Within this range of the Reynolds number, a shear-produced fully-developed turbulence has been produced in DNS. In the atmosphere, nonuniform wind is one of the primary sources of shear-produced fully-developed turbulence.*

31: Kolmogorov's theory has been reformulated for the atmosphere; see for example [5,6].

35-41: "Closure" betrays the real difficulty. The need for a bottom up, molecular approach has been largely ignored, but was pointed out in [1,2,3].

45-50: See references [7,8,9] for a discussion of Ri.

*The classical Kolmogorov theory of fully-developed turbulence was originally formulated for neutrally stratified homogeneous isotropic turbulence. Many turbulence closure models of stratified turbulence in meteorological applications have been based only on the density of the turbulent kinetic energy equation, without considering the evolution of the density of the turbulent potential energy proportional to the second moment of potential temperature fluctuations. In stable stratification, such turbulence closure models have led to the erroneous conclusion that shear-generated turbulence inevitably decays, and that the flow becomes laminar under "supercritical" stratifications (with the gradient Richardson number exceeding certain critical value). Contradictions of this conclusion, evidenced by the well-documented universal existence of turbulence under strongly supercritical conditions typical of the free atmosphere and the deep ocean, have been attributed to some unknown mechanisms and, in practical applications, handled heuristically.*

*We are uncertain about the feasibility of employing a molecular approach to describe velocity and temperature fluctuations in the inertial range of scales for fully-developed turbulence with large Reynolds numbers. Based on our current understanding, molecular simulations incur significant computational costs and may only be practical for small Reynolds numbers.*

60-79: The debate about dissipation ignores the reality of the Langevin equation and the approach to dissipation that emerges from a bottom up, molecular dynamics approach [4,10,11].

153 et seq: Couette flow has little physical reference to any atmospheric flow. The boundary conditions are far too restrictive. DNS is not direct. Molecular dynamics would qualify and is now almost within reach of current computational performance.

205: "dissipation time scale" is covering some major difficulties. See [4,10,11].

299-316: Dissipation is the process defining an operational temperature; it is infrared radiation to space.

*The effects discussed in Refs. [4,10,11], mentioned in the Referee report, are too different from those discussed in our paper. Contrary to Refs. [4,10,11], we consider a simpler system that does exclude humidity, radiation and photochemical effects, as well as phase transitions, cloud formation, and related physics. We have not studied complicated effects related to the intermittency of air temperature and its correlation with ozone photo-dissociation rate and the diurnal variation of ozone in the upper stratosphere.*

*We study a classical idealised problem of stably stratified shear-produced turbulence. In DNS, we use Couette flow for simplicity, because it allows us to*

*perform well-controlled numerical experiments. It assures a very certain fixed value of the Obukhov length scale, because in Couette flow the total (turbulent plus molecular) vertical fluxes of momentum and potential temperature are constant (i.e., they are independent of height). Our paper is relevant to the well-developed turbulence regime, where molecular transport is negligible compared to turbulent transport, so that turbulent fluxes practically coincide with total fluxes.*

**REFERENCES**

1. von Neumann, J. L. Recent Theories of Turbulence, in Collected Works of John von Neumann, Vol. 6, pp. 437-472, Macmillan, 1963.

2. Grad, H. Principles of the Kinetic Theory of Gases, in Handbuch der Physik, Band 12, S. Flugge, ed., pp. 205-294, Springer Verlag, Berlin, 1958.

3. Tuck, A. F. ATMOSPHERIC TURBULENCE: A Molecular Dynamics Perspective. Oxford University Press, Oxford, 2008.

4. Tuck, A. F. Scaling up: molecular to meteorological via symmetry breaking and statistical multifractality. Meteorology, 1, 4-28 (2022)

5. Schertzer, D.; Lovejoy, S. Physical modeling of rain and clouds by anisotropic scaling multiplicative processes. J. Geophys. Res. D, 92, 9693-9714 (1987).

6. Lovejoy, S.; Schertzer, D. The Weather and Climate: Emergent Laws and Multifractal Cascades. Cambridge University Press, Cambridge, UK. (2013).

7. Lovejoy, S.; Tuck, A. F.; Hovde, S. J.; Schertzer, D. Is isotropic turbulence relevant in the atmosphere? Geophys. Res. Lett. 34, L15082 (2007).

8. Lovejoy, S.; Tuck, A. F.; Hovde, S. J.; Schertzer, D. Do stable atmospheric layers exist? Geophys. Res. Lett. 35, L01802 (2008).

9. Hovde, S. J.; Tuck, A. F.; Lovejoy, S.; Schertzer, D. Vertical scaling of temperature, wind and humidity fluctuations: dropsondes from 13 km to the surface of the Pacific Ocean. Int. J. Remote Sensing 32, 5891-5918 (2011).

10. Tuck, A. F. Turbulence: vertical shear of the horizontal wind, jet streams, symmetry breaking and Gibbs free energy. Atmosphere, 12, 1414 (2021).

11. Tuck, A. F. Air temperature intermittency and photofragment excitation. Meteorology, 2, 445-463 (2023).

**RECOMMENDATION**

If the journal wishes to continue the largely unsuccessful grappling with turbulence that has characterized the last two centuries, then publish this paper -

it is better than most of the genre. But if so, the authors should acknowledge, however briefly, some of the difficulties outlined above.

---

## Author Response (AR2)

Dear Christian Franzke,

Thank you for your detailed feedback and for the opportunity to revise our manuscript. We sincerely appreciate the insightful comments from the reviewers and have carefully addressed each point raised in your letter. Below, we outline our responses to the specific issues discussed.

It is correct that z/L and Ri have equal rights as stability parameters. We prefer z/L over Ri because it is more convenient and provides simpler approximations. We explicitly state in Lines 237-239 that we exclude the underdeveloped turbulence of the viscous sublayer near the domain boundaries (plates) from our analysis. It is important to mention that DNS experiments effectively resolve turbulence even in close proximity to the domain boundaries. However, we exclude it because the goal of this study is to study well-developed turbulence, as stated in Line 170.

Regarding the observed spread of points, it is important to note that while the fully-developed steady state was achieved (verified using the standard criterion of stabilised TKE, which showed no significant fluctuations over time), the parameters involving theta' required additional time to stabilise. We believe that increasing the simulation time would decrease the spread. We have included this clarification in Lines 272-277.

To study intermittency, one needs to determine higher-order two-point (or multi-point) moments. Intermittency is important for small-scale effects, and intermittency implies that higher-order moments of velocity and temperature fields have non-Gaussian statistics. In our paper we focus on larger scales determining one-point second-order correlation functions barely touching one-point third-order correlation functions only when it is necessary. It also means, we are not in a position to make strong statements about intermittency. This clarification has been added in Lines 338-343 of the manuscript.

The figures have been redrawn to enhance clarity, showing only every 6th data point and representing the viscous sublayer using dotted light lines.

The concluding section has been restructured to emphasise the conclusions drawn from this study, summarise the achieved results, outline the remaining issues, and discuss future perspectives.

The minor remarks were also addressed in the revised manuscript.

We hope these revisions effectively address the concerns raised by the reviewers. We are thankful for their constructive critiques, which have undoubtedly enhanced the quality of our work.

Best regards,
On behalf of all co-authors,
Evgeny Kadantsev